# A Scoping Review of Educational Interventions to Increase Prosociality against Gender-Based Violence in University Bystanders

Maria Villalonga-Aragón [1], Manuel Martí-Vilar [1,*], César Merino-Soto [2,*] and Lizley Tantalean-Terrones [3]

1 Departamento de Psicología Básica, Universitat de València, Avgda. Blasco Ibañez 21, 46010 Valencia, Spain; mariavillalonga4@gmail.com
2 Instituto de Investigación de Psicología, Universidad San Martín de Porres, Lima 34, Peru
3 Facultad de Psicología, Universidad Nacional Federico Villarreal, Lima 15088, Peru; ltantalean@unfv.edu.pe
* Correspondence: manuel.marti-vilar@uv.es (M.M.-V.); sikayax@yahoo.com.ar (C.M.-S.)

**Abstract:** Gender-based violence represents a problem of public interest with a high prevalence on university campuses, which has intensified the preventive strategy for potential victims. However, the prosocial action of the viewer provides a promising alternative to mitigate its incidence. This study analyzes preventive interventions based on gender-based violence towards university viewers as a basis for future proven and sensitive implementation processes of intercultural adaptation. The Scoping review process was implemented based on the Arksey and O'Malley methodology (aligned with the Cochrane manual), with subsequent reporting of results according to the PRISMA guide. In total, 15 articles finally met the selection criteria in the databases: WOS, Scopus, ERIC, PsycINFO, Embase, and PubMed, with specific descriptors. Despite the heterogeneity in the types of intervention and the study variables, the intervention and the results are described, and it is highlighted that they were effective for most of the proposed objectives, such as the decrease in the perpetration of sexual violence, the increase in prosocial behavior, and the recognition of forms of violence. The great value of these preventive interventions is concluded, and the diversity of these strategies implemented in parallel and continuously would have a lasting impact in higher education contexts.

**Keywords:** prevention; intervention; gender-based violence; intimate partner violence; programs; university students; bystanders; Scoping review

## 1. Introduction

Individuals' and communities' well-being loss is even more multifactorial in diverse conditions, where minorities are vulnerable to having their health and security requirements disregarded (Costa 2023). In addition, community spaces with high rates of diversity are exposed to a lack of interpersonal identification and communication shortages. That enhances the presence of frustration and negative feelings as the origin of violent behavior (Baron and Neuman 1996). In this sense, University campuses, as a global tendency, are likely to be vulnerable to different expressions of violence, so their minorities require prosocial support that enhances their well-being.

### 1.1. Bystander Behavior in Social Psychology

During the 20th century, research in social psychology has been mainly motivated by the predictability of behavior based on mediating processes. Therefore, much of the scientific production of the last century have tried to identify, isolate, describe, and explain concepts related to thought and action (Montero 1994).

Social psychology showed curiosity and concern for the inaction of human groups in the face of emergent events, such as what happened in 1964 with CSG, a 28-year-old girl who was the victim of an assault with fatal consequences after not receiving help in time.

Thirty-eight neighbors witnessed the event, but only one person called the police when the aggressor left the place (Gangsberg 1964). Although the eventuality does not show the number of witnesses or their degree of exposure to the event, various authors focus on a new conceptualization of group behavior (Manning et al. 2007). Under this premise, a bystander is a person present in each scenario who possesses the power to assume or avoid responsibility for its action.

As a prelude to the 1970s, the social psychologists Bibb Latané and John Darley promoted the first experimental study on the bystander, regarding his inhibition of action or transfer of responsibility (Levine 2012), based on behavioral predictability associated with the subject's attitude, as an effect of the adjustment in his subjective perception of the norms of "pass" and "subjective control" and even his "value-expectation" belief scheme (Ajzen 1991). The experiments of Darley and Latané (Darley and Latane 1968) describe and explain the behavior of the spectator, who, regardless of his personality type, can imitate or influence the behaviors of others, assuming a proactive, apathetic, alienated, or anomic attitude. For this purpose, they posed emergent cases in experimental conditions, proving that the greater the presence of bystanders, the less supportive or helpful intervention was in the experimental subjects (Myers 1983). From this experience, the five moments in the bystander's prosocial intervention are: (a) identifying the situation; (b) perceiving the risk in potential victims; (c) assuming the responsibility to intervene; (d) deciding to act; and (e) acting (Latané and Darley 1968, 1970). Burn (2008) applied this model to sexual and gender-based violence from the bystander perspective, identifying the five inhibitory barriers to prosocial action. These barriers involve the diffusion of responsibility (Levine 2012) and the determination of the victim (as potential or actual) in terms of their observable characteristics of appearance and behavior (Burn 2008). In this sense, the importance of understanding the behavior of the university spectator in relation to the approach of dating violence in their peer group is highlighted (Cusano et al. 2020).

*1.2. Gender-Based Violence on University Campuses and Bystander Intervention*

According to the World Health Organization (WHO), gender-based violence constitutes a human rights abuse and a public health problem of epidemic proportions (WHO 2013), with a need for urgent action, as indicated in the World Report on Violence and Health of the Pan American Health Organization (PAHO) (Pan American Health Organization 2002). Data from 21 US universities surveyed in 2015 indicate that 1 in 5 female students and 1 in 20 male students experienced sexual assault during their formative years (Association of American Universities 2015). In the following four years, despite efforts to implement preventive programs and increased resources to identify this problem among students, the prevalence of sexual assault cases remained stable, and the rate of sexual assault persisted (Association of American Universities 2019; Jozkowski 2015). Therefore, the university population requires psychoeducation on sexual consent based on sociocultural factors (Ortiz and Shafer 2018).

Sexual violence prevention programs on college campuses generally address the establishment of limits on sexual consent for women (Berkowitz 2002; Davis 2000; Fabiano et al. 2003). They also include psychoeducational proposals for developing positive and proactive attitudes in the male population (Flood 2006; Gibson 2014). However, recent publications (Chabot et al. 2016; Deitch-Stackhouse et al. 2015) reflect that bystander intervention overcomes the dualistic view of sexual assault prevention focused on the victim and the aggressor (Mennicke et al. 2018; Brown et al. 2014; Lemay et al. 2019; Palmer 2016) and focuses on eradicating environments tolerant of violence and sexual assault. This model involves the inclusion of community intervention strategies for the promotion of socio-normative changes (Banyard et al. 2003). The prosocial model of the bystander is included (Darley and Latane 1968; Banyard et al. 2014, 2007; Crooks et al. 2018), for their empowerment in the face of situations at risk of sexual violence. These interventions aim to teach the college student to recognize the warning signs of possible sexual assault and encourage them to intervene if they see a peer in distress (McMahon and Banyard

2011). Some of these experiences increased bystander prosocial behavior (Banyard et al. 2007; Coker et al. 2011), but others did not (Gidycz et al. 2011). Regardless of their results, short-duration, group-focused program formats are identified as having limited overall reach and impact (Salazar et al. 2019).

Prosocial behavior, a differential construct of altruism, is associated with normative judgments, social skills, and self-regulatory capacity (Caprara et al. 2005). It acquires vital importance due to its association with the sense of community, attention, and care of the group to which it belongs, especially when emergencies arise (Martí-Vilar et al. 2019). This sense of community promotes action to interrupt aggression or a situation of potential aggression, going against social norms that support sexual violence, and being an effective and supportive ally for survivors (Cares et al. 2014).

Nevertheless, most of the primary prevention research in this field comes from the North American context. A cross-national review of violence intervention and prevention programs (Ellsberg et al. 2015) revealed that more than 80% of the rigorous evaluations were conducted in six high-income countries, which account for 6% of the world's total population.

In this topic, systematic reviews and meta-analyses of programs that address sexual violence among university students were identified. In this regard, Park and Kim (2022) analyzed the effect of programs to reduce the risk of intimate partner violence in 13 randomized controlled trials. In turn, Finnie et al. (2021) included 28 studies in their review, considering interventions in intimate partner violence based on the promotion of alternative behaviors to aggression and the creation of protective environments. Additionally, Evans et al. (2019) analyzed 11 intervention studies on university bystanders, diverse in duration, instrumentation, and educational strategy. Likewise, Mahoney et al. (2019) applied the Haddon Matrix to evaluate intervention programs against sexual aggression on university campuses, selecting 31 articles published between 2001 and 2017, including experimental and quasi-experimental design programs. Finally, Jouriles et al. (2018) analyzed 24 articles on bystander attitudes, beliefs, and behaviors, their effects, and sustainability.

It should be noted that, although the mentioned studies are systematic reviews on educational programs against gender violence, they present certain limitations and parameters: three articles include publications on primary preventive intervention in intimate partners and tertiary preventive intervention (towards the perpetrator victim dyad), not being exclusive on bystanders (Park and Kim 2022; Finnie et al. 2021); also the presentation of the assessment and analysis of their effects (Finnie et al. 2021; Evans et al. 2019); the selection of experimental design programs in randomized controlled groups (Park and Kim 2022; Mahoney et al. 2019); and finally, the analysis of descriptive studies and processes of promotional and preventive intervention in bystanders (Jouriles et al. 2018). The main contribution of the present review study is that each selected article is based on educational interventions in prosocial behavior applied to spectators (as the main target population), regardless of their scope, methodology, intervention strategy, or evaluation of their sustainability over time.

Review articles contribute to scientific dissemination by synthesizing the state of research in various disciplines and themes and constituting a starting point for research thanks to the integration and updating of results (Fernández-Ríos and Buela-Casal 2009). The relevance of a Scoping review based on violence prevention programs allows a realistic analysis of the most used strategies and the revalidation of effective methodologies, which is very useful for formulating public policies and implementing social programs (Provost et al. 2021).

Therefore, the main objective of the research was to systematize the studies based on university bystander education programs for the prevention and intervention of gender-based violence and to carry out a descriptive study. For this purpose, we aimed to contrast the studies based on the interventions implemented and executed, their research design, the type and size of the sample, the sociodemographic characteristics of the participants, the measurement instruments, and the results of the effect or effectiveness of the interventions.

## 2. Materials and Method

### 2.1. Protocol

The present study evidences a Scoping review of the scientific literature published in preventive-prosocial intervention programs directed to mitigate gender-based violence in university contexts (Appendix A). Its elaboration includes the question, the inclusion-exclusion criteria, the review and selection of studies, the extraction of data with their respective analyses and the report of results. In confluence, the Cochrane review regarding the effects of health or social interventions (including randomized and non-randomized studies) (Higgins et al. 2022). Finally, these findings are reported in terms of the PRISMA statement (Yepes-Nuñez et al. 2021) for the publication of Scoping reviews (Figure 1).

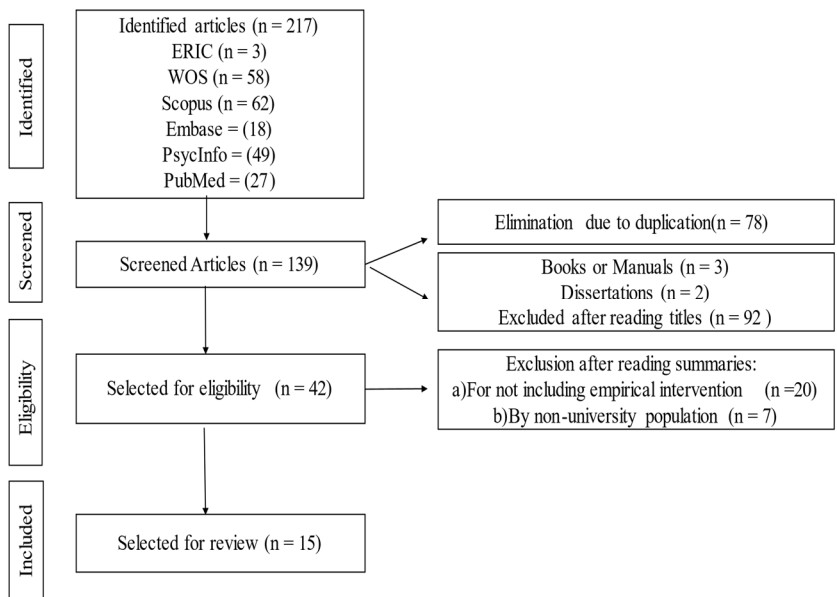

**Figure 1.** PRISMA diagram, study selection process. Preferred Reporting Items for Systematic Reviews and Flowchart of Meta-Analyses (PRISMA).

### 2.2. Search

This bibliographic search took place between August and September 2021. It was carried out in different social and cultural contexts, referring to educational interventions implemented in the USA, UK, Vietnam, Italy, and South Korea; the registration of the country was important to increase the generalization of the results. The search was confirmed in three phases. First of all, a search to obtain an overview of the subject in question was carried out; secondly, the application of the inclusion and exclusion criteria to delimit and centralize the topic was performed; and, finally, a manual search to include those articles that had not been found after the first itinerary was carried out.

The search strategies allowed us to identify the maximum number of studies on bystander education programs for university students in the face of gender-based violence. The review was conducted in English and Spanish, under the combination of two terms joined with the connector [AND], with no limit on the year of publication. Likewise, the manual search was performed by checking the list of references of the articles that were significantly relevant to the study, in order to access potentially additional studies.

The initial question (PICO) that guided this review was built as follows: Do educational interventions increase prosociality in the face of gender violence among university bystanders? University bystanders are considered the participants (P), the interventions (I) are educational, comparisons (C) are either made with pre- and post-test measurements, between-group comparison, or a control group, and outcomes (O) try to measure the effects that the programs have on gender violence.

*2.3. Eligibility*

A protocol was registered in PROSPERO and the search was performed according to the following criteria. The identification code is CRD42022337692.

2.3.1. Inclusion Criteria

The included studies conform to the following formats: (a) Articles referring to bystander intervention processes in gender-based violence; (b) Studies of an experimental nature (whether pre-experimental, quasi-experimental, or controlled experimental); (c) Studies involving modeling effects in viewer behavior; (d) Research that includes a university population, regardless of gender, ethnicity, geographic location, educational level, or age range; and (d) Studies published in the English language.

2.3.2. Exclusion Criteria

Studies were discriminated in the following formats: (a) Articles associated with social experiments (studies of a psychological and sociological nature, with exposure of the participants to specific stimuli and consequent immediate measurement), since they do not correspond to structured processes of preventive intervention; (b) non-experimental descriptive-comparative, cross-sectional studies that collect information on the subject but do not record the manipulation of stimuli for learning alternative behaviors to cope with gender violence; (c) Articles on content validation of gender-based violence prevention programs; (d) Books, manuals or dissertations, (e) Thesis of all kinds (degree or professional qualification), as they are not exposed to the scrutiny of a scientific journal; (f) Editorials; (g) Clinical case studies and (h) Exclusion of gray literature with the use of secondary databases such as Google Scholar.

In the process of identifying the studies, the following terms were considered: Gender-based violence AND prosocial* AND intervention, and Intimate partner violence AND prosocial* AND intervention, as combinations indicated for the databases: Web of Science (WOS), Scopus, of proven quality in their records (Cavacini 2014), Education Resource Information Center (ERIC), and PsycInfo. The combination used in PubMed was Intimate adheres partner violence AND prevention AND undergraduate students. Finally, in the case of Embase, the following was considered: Gender-based violence AND prosocial* AND intervention, gender-based violence AND bystander AND intervention; and partner violence AND prosocial* AND intervention. It should be noted that the last four databases are included according to the objectives of the review associated with the fields of higher education, mental health, and the addition of topics of psychosocial interest.

*2.4. Data Collection*

Once the articles were obtained, they were exported to Covidence, a screening and data extraction tool for conducting reviews of standard interventions. This resource allows duplicate detection and allows each reviewer to examine the titles and abstracts of the articles and decide whether to include or exclude them by applying the previously established criteria. In case of a discrepancy, the arguments were discussed together. Next, the selected documents were recorded in the Mendeley bibliographic manager and an Excel spreadsheet. To systematically analyze the information contained in the articles, the following categories were completed: (a) authors, year of publication, and country; (b) type of intervention; (c) methodology used and the presence or absence of a control group; (d) sociodemographic characteristics of the sample; (e) the study variables; (f) the evaluation instruments used; and (g) the results obtained.

*2.5. Selection*

The resulting search targeted 217 articles: 3 in ERIC, 58 in WoS, 62 in Scopus, 18 in Embase, 49 in PsycInfo, and 27 in PubMed. The papers were downloaded in a text file and subsequently analyzed based on the title, abstract, keywords, type of intervention, and results. As a result of this first evaluation, 78 duplicate articles were removed and 92 studies

were discarded in the first screening after reading the title. Then, 42 articles were preselected and their full text read. Of these, 39 articles were selected, and after applying the inclusion and exclusion criteria, 27 articles were eliminated from the review. At the end, a final selection of 15 articles was included in this Scoping review, whose paramount requirement was to consider the modification of viewer behavior through stimuli or psychoeducational processes, which imply results of effect or effectiveness (Figure 1).

Two researchers carried out the final selection of the studies included (M.V-A. and M.M.-V.). Their decisions were based on the methodological quality they presented, following the SQUIRE Guidelines 2.0. After that, three researchers (M.V.-A., L.T.-T. and C.M.-S.) were responsible for assessing them according to three categories related to their quality (low, medium, high). Only those articles considered of high quality were included. Once the process was finished, none of them were excluded.

### 2.6. Synthesis of Results

The synthesis of the results was formulated based on six pre-established criteria, that integrate common aspects of the studies reviewed: (a) Type of intervention; (b) Design of the educational intervention; (c) Type of sample and its distribution; (d) Sample size; (e) Demographic characteristics of the sample; (f) Measuring instruments; and (g) The main results of the studies. These criteria are observed in the results section (See Table 1) and are the subject of analysis in the discussion.

**Table 1.** Articles that reflect the intervention in the prevention of gender-based violence in bystanders.

| Year, Authors, Country, and Citations | Kind of Intervention | Objectives | Design/ Sample | Sample Distribution | Instruments | Results |
|---|---|---|---|---|---|---|
| Kuffel & Katz 2002 USA *PsycInfo* 54 Citations (Kuffel and Katz 2002) | Educational video followed by a discussion led by a male/female presenter or co-presenters. The Control group viewed an episode of the series "Friends." | To assess the effectiveness of a brief program to prevent physical, psychological, and sexual aggression in intimate relationships. | **Design:** Intervention Group with Control Group. **Sampling:** Random | N = 123 EU (Initial), n = 76 EU (Final) IG: 36.8% (n = 28)—CG: 63.2% (n = 48) Women 63.4%, Men 36.6% | 1. Relationship Expectations Scale (RES; Washington & Fiore, 1997), 2. Scenarios to discriminate physical, psychological, and sexual abuse; aggressive and abusive dating behavior. 3. Conflict Tactics Scale-Revised (CTS-2); attitude towards the program. | The program is effective for the IG with: • Improved identification of types of abuse. • Decreased aggressive attitudes. • The results were not sustainable |
| Coker et al. (2014) The United Kingdom *Scopus* 150 Citations (Coker et al. 2014) | Green Dot—Bystander training program that engages the US in actions to reduce SV | To compare the rates of the types of violence in the US before and after the intervention in the IG and CG. | **Design:** Experimental Comparative Observational **Sampling:** Stratified Random | N = 7026 EU (From three campuses) IG: 39.4% (n = 2768) CG: 60.6% (n = 4258) IG: Men (49.6%), women (50.4%), CG: Men (47.5%), women (52.5%) GI: Caucasian (79.4%), Afro-descendant and others (20.6%); CG: Caucasian (77.5%), Afro-descendant and others (22.5%) | 1. Victimization and violent perpetration—Adaptation of (a) Forced or unwanted sex (NSSIPV), (b) Sexual Harassment (QES, by Payne et al.), (c) Stalking (NSVAW), and (d) Physical and Psychological Violence in courtship (RCTS). 2. Sociodemographic attributes | The program: • Decreased rate of victimization and perpetration of SV. |
| Potter et al., 2008 USA *PubMed* 29 Citations (Potter et al. 2008) | Media campaign around four posters with a modeling effect on alternative behaviors in a VAW risk situation | Analyze the role of campaigns in addressing a public health problem. Describe the implementation and evaluation of the strategy in the reduction of incidences of VAW | **Design:** Quasi-Experimental with CG (exploratory). **Sampling:** Intentional | N = 145 EU IG: 55.9% (n = 81)—CG: 44.1% (n = 64) Women 51%, Men 49% | 1. Knowledge of means of intervention in case of VAW. 2. Perception of learning and assessment of the program post-intervention) 3. Development of focus groups to improve the content of visual stimuli. | The campaign: • Validated its content with the intervention of the participants. • Increased prosocial behaviors. |
| Potter et al., 2011 USA *PubMed* 54 Citations (Potter et al. 2011) | "Know your Power": Poster campaign for the prevention of sexual violence, under the concept of social self-identification | Evaluate the effectiveness of the posters and their internalization for the intervention as a prosocial bystander. | **Design:** Quasi-experimental 01 measure **Sampling:** Voluntary (External motivation [EM]) | They viewed the posters: (n = 291) EU; They did not view the posters; (n = 81) Completed survey: (n = 372) USA Women 61%, Men 39% Caucasian (87%), other (13%) | 1. Willingness to participate in preventing sexual assault 2. Social self-identification as witnesses in VAW cases. | The program: • Was effective in the phases of contemplation and action • It had a greater impact on women's actions. |
| Salazar et al., 2014 USA *WOS* 112 Citations (Salazar et al. 2014) | RealConsent: Web-based general health promotion program or comparison (control) program. | Check the effectiveness of the Program to prevent the perpetration of SV, increase prosocial behavior, and learn about the relationships of theoretical mediators in the US | **Design:** Experimental with CG **Sampling:** Random | n = 743 EU (Baseline) n = 451 EU (Post Intervention) n = 215 (Follow-up) IG: 51% (n = 376) CG: 49% (n = 367). Heterosexual or bisexual men Caucasian (44%), African American (22%), Asian (20%), Latinos (11%) and others (3%) | 1. Primary Measures: (a) Reactions to Offensive Language and Behavior (ROLB) (b) Conflict Tactics-II (CTS2), 07 items. 2. Questionnaire on theoretical mediators of implied consent. | The program • Decreased DV perpetration and increased PI, but did not trigger bystander intervention. • Promoted significant differences in the 12 mediators evaluated ($p < 0.001$) |
| Cares et al., 2015 USA *WOS* 80 Citations (Cares et al. 2014) | Bringing in the Bystander: 2-session program (only with the IG) Know your Power": Social marketing campaign with modeling and reinforcement effect for GI and GC (Posters with scenarios that address VS incidents). | Evaluate the effectiveness of the program and the campaign in two university fields: (a) Pre and post Intervention, (b) Follow-up I (5 months later), (c) Follow-up II, and (12 months later), parallel to the sociodemographic contrast | **Design:** Experimental with GC **Sampling:** Voluntary EM | n = 948 EU (Baseline) n = 607 EU (Post Intervention) n = 346 EU (Follow-up) IG: 49% (n = 466) CG: 51% (n = 482) Pre-Intervention: Men (51.5%), women (47.8%), and 03 transgender participants. Caucasian (73.2%), Afro-descendant (26.8) | 1. Social desirability (Stöber 2001) 2. Adapted Illinois Rape Myth Acceptance (Payne et al. 1999) 3. Malamuth's Attraction to Sexual Violence 4. Bystander Efficacy (Banyard 2008; Banyard et al. 2007) 5. Intention to Help a Friend and Stranger Scale (Banyard et al. 2014) 6. Knowledge Items (2nd edition) (Banyard et al. 2007). | The Program is effective: • Female students on campuses with a larger male population. • For prosocial action in the face of an SV situation. |
| Senn & Forest, 2016 USA *PsycInfo* 31 Citations (Senn and Forrest 2016) | Bringing in the Bystander: Workshops on bystander intervention against the social norms that validate AS and coercion. Promotes strategies of recognition and safe interruption of potential SA events. | To evaluate the efficacy of education for preventing Sexual assault in bystanders (developed by undergraduate students trained for this purpose). | **Design:** Quasi-Experimental Single group. **Sampling:** Voluntary EM | n = 827 EU (Initial) n = 444 (Final) (1) Initial Sampling—IG: 62.6% (n = 518)—CG: 37.4% (n = 309), (2) Final Sampling—IG: 56% (n = 248)—CG: 44% (n = 196) Women 78.3%, Men 20.7%, other genders 1% Caucasian 96.1%, Other African American—Caribbean, Asian or Middle Eastern 3.9% | 1. Bystander efficacy (Banyard et al. 2014) 2. Preparation change/Help (Banyard et al. 2014), 3. Brief viewer intent (Banyard et al. 2014), 4. Barriers to intervention for sexual assault (Burn 2008) 5. Viewer behavior (adapted from Banyard et al. 2007, in Banyard et al. 2014). 6. Social Desirability Scale—SDS-17 (Stöber 2001). 7. Careless Response | The program: • Increased willingness toward proactive intervention, independent of third-party criticism. • Decreased pre-contemplative beliefs. • The effects lasted four months. |

**Table 1.** *Cont.*

| Year, Authors, Country, and Citations | Kind of Intervention | Objectives | Design/ Sample | Sample Distribution | Instruments | Results |
|---|---|---|---|---|---|---|
| Mennicke et al., 2018 USA *PsycInfo* 15 Citations (Mennicke et al. 2018) | Social norms marketing campaign to engage men in AS prevention (Implemented for five years) | Evaluate the campaign's impact on positive and prosocial attitudes, beliefs, and behaviors towards SA and bystander intervention with the change of norms on campus. | **Design:** Quasi-Experimental of a single group. **Sampling:** Random | 3000 EU for each annual period N = 15,000 EU based on initial sampling and n = 4158 participants over 5 years Heterosexual men 90.9%, bisexuals 2.5%, Gays 5.4% Caucasian 68.5%, Hispanic 16.5%, Asian 3.6%, African American 28% and 6.4%, Other 4%. | 1. Multidimensional survey based on attitudes: (a) Beliefs in rape myths (Illinois Scale—11-item Adaptation (Payne et al. 1999), (b) Sexually aggressive behaviors, and (c) Bystander intervention behavior (six items), with self-assessment and co-assessment processes. | The social norms marketing campaign: • Positively affected the perceptions, attitudes, beliefs, and self-reported prosocial behaviors related to acts of sexual assault. |
| Ortiz & Shaffer, 2018 USA *Scopus* 9 Citations (Ortiz and Shafer 2018) | "Define Your Line": Campaign to "unmask the lines" of sexual consent through peer-to-peer and mediated messaging. | To test the effectiveness of the education campaign on sexual consent promoted by university students. | **Design:** Single group quasi-experimental (exploratory). **Sampling:** Voluntary EM | N = 992 US. Pretest (n = 324), Posttest 1 (n = 328), Posttest 2 (n = 340) Women 60.3%, Men 39.7%, Others 1.4% Caucasian (63.2%), Mixed Race (7.4%), Latin American (24%), African American (4.8%), Asian (4.3%) | 1. Adaptation of the Humphreys and Brousseau. Revised Sexual Consent Scale 2. Behavioral control scale (8 elaborated items) 3. Adaptation of the Sexual Consent Request Behaviors Scale 4. Scale to determine whether or not it is a sexual assault (previous presentation of 12 analysis situations) | The campaign: • Improves understanding of sexual consent, as a function of exposure time. • Has greater effectiveness in males who belong to fraternities. |
| Salazar et al., 2019 USA *WOS* 7 Citations (Salazar et al. 2019) | RealConsent: Web-based sexual violence prevention program or Control Condition | Check the effectiveness of the Program to prevent the perpetration of SV, and increase the prosocial behavior of the viewer and the validity of theoretical models | **Design:** Quasi-Experimental with GC **Sampling:** Random | N = 743 EU IG: 51% (n = 376) CG: 49% (n = 367) Heterosexual or bisexual men. Caucasian (44%), African American (22%), Asian (20%), Latino (11%), and Other (3%) | 1. Reactions to Offensive Language and Behavior (ROLB). 2. Conflict Tactics-II (CTS2), 07 items- | The program is effective with: • Decreases perpetration in SV to 28% • Increases prosocial behavior to 12%, but is not associated with bystander action. Validates theoretical models correlational. |
| Cusano et al., 2020 USA *WOS* 2 Citations (Cusano et al. 2020) | Focus groups (GF), 60 to 75 min on dating violence (DV) perspectives. | To analyze the understanding and knowledge of DV and its impact on the decision to intervene as a prosocial bystander. | **Design:** Qualitative Grounded theory **Sampling:** Voluntary knowledge and interest | N = 43 EU 03 G of Men, 04 G of women, and 02 G. mixed Men (= 40%), women (n = 51%), and OG (n = 9%) Caucasian (42%), Asian (23%), African American (28%), and Latino (26%) | 1. DV structured protocol (thoughts, definition, perception of victim support, knowledge of policies, campus resources, experiences with those involved in abusive relationships, and prosocial intervention predisposition. | The Focus Groups reflect prevention: • Recognition of unhealthy and abusive behaviors • Ability to determine risks. • Discrimination in emergencies |
| Yount et al., 2020 Vietnam *WOS* 6 Citations (Yount et al. 2020) | GlobalConsent—Program adapted from RealConsent for use in Vietnam or Web-based Control Educational Program. | To test the impact of the adapted program (GlobalConsent) in preventing sexual violence and prosocial behavior of university viewers. | **Design:** Experimental with CG. **Sampling:** Random | N = 793 EU (n = 345 UPV and n = 448 PU) IG: 50% (n = 397) CG: 50% (n = 396). Heterosexual or bisexual men does not indicate ethnicity. | 1. Questionnaire on family history, exposure to violence in childhood, and other forms of sexual content. 2. RealConsent evaluation forms and scales (adapted) to GlobalConsent modules. | • The program is effective in the Vietnamese population and for adapting to other Low and Middle-Income countries. |
| Santacrose, L.B.; Laura, A.C.; Marchell, T.C. 2020 USA *PsycInfo* 5 Citations (Santacrose et al. 2019) | Intervene a 20-min video, with the effect of modeling the viewer's prosocial behavior, with strategies to intervene in 7 situations. | To assess the effectiveness of the video in increasing the self-reported probability of intervention (pre and post-video viewing) with a 4-week follow-up. | **Design:** Experimental with CG. **Sampling:** Random | n = 1243 EU (Initial), n = 853 (Final) 1) Initial Sampling—IG: 35.7% (n = 444)—CG: 64.3% (n = 799), 2) Final Sampling—IG: 40.3% (n = 344)—CG: 59.7% (n = 509) Women 55.9%, Men 44.1% | 1. Survey around the bystander model (5 steps), three questions around the prosocial model, and two about social norms in twelve different situations, seven of which focused on intervention. | • The video increased the probability of prosocial intervention, in favor of women, in cases of: (a) intimate partner violence, (b) sexual aggression, and c) sexual harassment. |

**Table 1.** *Cont.*

| Year, Authors, Country, and Citations | Kind of Intervention | Objectives | Design/ Sample | Sample Distribution | Instruments | Results |
|---|---|---|---|---|---|---|
| Martini, M.; De Piccoli, N. 2020 Italy *EMBASE* 0 Citations (Martini and Piccoli 2020) | USVreact: a 4-month training program for university staff to counter sexual violence | Evaluate the program's effectiveness regarding the gender system, the identification and evaluation of SV risks, and the predisposition to intervention. | **Design:** Quasi-experimental single group <br><br> **Sampling:** Voluntary EM | N = 66 (02 university communities) 87.5% staff, 9.4% teachers, 1.6% university leaders and 1.6% directors) Men (7.5%), women (92.5%) Does not indicate ethnicity | 1. Justification of the gender system (Jost and Kay 2005) 2. ESV, an adaptation of the Fitzgerald Scale 3. Subtle rape myth from McMahon and Farmer (2011) in its Italian adaptation (SRMA-IT) by Martini et al. (2021). 4. Bystander intention to intervene (Banyard et al. 2003, 2005) | The program: <br> • Increases recognition of subtle forms of violence. <br> • Reduces acceptance of the rape myth. <br> • Develops active and reactive attitudes in situations of SV. <br><br> Depending on the supportive reaction culture. |
| Park, S.; Kim, S.H. 2021 South Korea *EMBASE* 0 Citations (Park and Kim 2021) | With You Education: Bystander Program | Evaluate the acceptance and impact of the program (designed to improve the skills of friends who support DV Victims | **Design:** Quasi-Experimental Single Group <br><br> **Sampling:** Voluntary EM | N = 46 EU Men (24%), women (76%) Does not indicate ethnicity | 1. Competence of supporters to help victim friends, with eight scales from the Physician Readiness to Manage intimate partners Violence (PREMIS). 2. Satisfaction questionnaire used in (E. Park et al. 2017, p. 200). 3. Focus group to identify participants perceptions. | The program is effective in: <br> • The willingness to help. <br> • The understanding of victims, and their awareness of DV as a public health problem. |

Sexual Experience Questionnaire (QES), National Survey of Sexual and Intimate Partner Violence (NSSIPV), University Students (EU), Intervention Group (IG), Control Group (GC), Prosocial Intervention (PI), Domestic Violence (DV), Experience of Sexual Violence (ESV), National Survey on Violence against Women (NSVAW), Violence against Women (VAW), "Revised" Conflict Tactics Scale (RCTS).

## 3. Results

A total of 15 documents met the criteria to be part of the review. Table 1 shows a synthesis of all of them. A brief description follows.

All the analyzed articles were published in the last 19 years, 86.7% (n = 13) in the previous 10 years, and 69.2% (n = 9) in the last 5 years (review, the year 2021 has not yet ended and the number may have increased). Additionally, 73.3% (n = 11) of the research has been carried out in the USA (Cusano et al. 2020; Ortiz and Shafer 2018; Mennicke et al. 2018; Salazar et al. 2014, 2019; Cares et al. 2014; Cavacini 2014; Coker et al. 2014; Potter et al. 2011; Senn and Forrest 2016; Santacrose et al. 2019). The remaining percentage has been performed in the United Kingdom (Coker et al. 2014), Vietnam (Yount et al. 2020), Italy (Martini and Piccoli 2020), and South Korea (Park and Kim 2021) (n = 1 for each country). The first published study (Kuffel and Katz 2002) has 39 citations on ResearchGate and 17 on Springer Link. The number of publications has been increasing, as can be seen. Likewise, Coker et al. (2014) and Salazar et al. (2014) studies stand out for their number of citations (with 150 and 112 mentions, respectively), in contrast to Martini et al. (2021) and Park and Kim (2021) studies. They did not report citations, probably due to the time of publication, the novelty of the type of preventive intervention, and the accessibility of the means of publication.

### 3.1. Types of Intervention

In this regard, a study based on focus groups of 60 to 75 min assessed students' experiences, behaviors, and attitudes toward gender-based dating violence (Cusano et al. 2020). Three articles on web-based intervention programs (Salazar et al. 2014, 2019; Yount et al. 2020); two of them (Salazar et al. 2014, 2019) implemented RealConsent (the only sexual violence prevention program for college students), which is considered effective according to the Centers for Disease Control and Prevention (Salazar et al. 2014; Dills et al. 2016). The third article implements and executes the adaptation of the RealConsent program in the Vietnamese population, determining said version as Global Consent, based on the preventive approach to sexual violence and promoting prosocial behavior (Yount et al. 2020).

Likewise, some intervention programs were implemented on campus (Cares et al. 2014; Coker et al. 2014; Senn and Forrest 2016; Park and Kim 2021) and through social marketing campaigns (Ortiz and Shafer 2018; Mennicke et al. 2018; Cares et al. 2014; Potter et al. 2008, 2011). As well as the combination of both strategies: the implementation of an intervention program (Bringing in the Bystander), followed by a social marketing campaign called "Know You Power" (Cares et al. 2014). This campaign was also executed under the concept of social self-identification (Potter et al. 2011) through discussion groups and surveys, with focus groups and surveys associated with the context of the target audience. From another angle, the "USVReact" (Martini and Piccoli 2020) program involved administrative personnel who work on university campuses because their permanence transcends the passing of different generations of students and their contribution to the construction of an organizational and educational environment can be stable and sustainable over time. In addition, the "Bringing in the Bystander" program (Senn and Forrest 2016) is considered highly effective for assessing the pronouncement against social norms that support sexual aggression or coercion and for identifying and safely interrupting those situations that can lead to sexual violence. These facts become an effective and supportive ally with rape survivors (Banyard et al. 2007). The media campaign from the multidisciplinary approach requires mention in which professors, university staff, and selected students participated (Potter et al. 2008). Along the same lines, the "Define You Line" (Ortiz and Shafer 2018) campaign was carried out with a community approach, thanks to the encouragement of university students and other faculty members.

These experiences include the exposure of the participants to 20-min educational videos, as is the case with the shorts "Intervene" (Santacrose et al. 2019) and "Choices"

(Kuffel and Katz 2002). The latter four segments represented physical, psychological, and sexual aggressions, followed by a guided discussion.

These processes are supported by different theories, among which stand out: Bandura's social cognitive theory (Bandura 2004; Salazar et al. 2019, Santacrose et al. 2019); Latané and Darley's situational model of viewer behavior (Latané and Darley 1969, 1970; Cusano et al. 2020; Park and Kim 2021); the social norms theory of Fabiano et al. (2003) and Perkins and Berkowitz (1986) (Mennicke et al. 2018; Salazar et al. 2019; Santacrose et al. 2019) and the transtheoretical model of Prochaska and Diclemente (1984) (Cares et al. 2014; Potter et al. 2011; Senn and Forrest 2016).

*3.2. Design*

The selected studies favor the implementation of experimental designs, with one exception (Cusano et al. 2020), which focused on developing focus groups on the understanding of dating violence and its impact on viewer behavior. The study provides the participants with a space for expression and reflection, from the subjective conception towards the achievement of joint conclusions; therefore, it is considered an "intervention" (before the learning that it implies). It also includes the delivery of preventive material at the end of each conversation.

Experimental studies (Kuffel and Katz 2002; Salazar et al. 2014; Yount et al. 2020; Santacrose et al. 2019) are the ones that denote greater control in the selection process of the matched sample and favor the creation of supervised conditions in the treatment of intervention groups and control groups. In these four studies, two particular conditions emerged: (a) The implementation of educational sessions (Salazar et al. 2014; Yount et al. 2020), in the American and Vietnamese contexts, respectively, based on the decrease in the perpetration of SV and the increase in prosocial behaviors; and (b) The presentation of educational videos with a modeling effect (Kuffel and Katz 2002; Santacrose et al. 2019). In the first case towards an increase in the probability of intervention and in the second case towards a preventive attitude of physical, psychological, and sexual violence.

Quasi-experimental studies are identified as those that follow the experimental model but do not achieve total surveillance of the intervening variables, including the sample selection process. Three studies included a control group (Salazar et al. 2019; Cares et al. 2014; Potter et al. 2008). This condition allows a comparative contrast that validates the effectiveness of the intervention process (as a similar effect). Five studies considered only the intervention group (Ortiz and Shafer 2018; Mennicke et al. 2018; Coker et al. 2014; Senn and Forrest 2016; Martini and Piccoli 2020; Park and Kim 2021), given the number of participants and the conditions of massive and indistinct intervention, which is understandable when it comes to long-term campaigns or programs.

In this context, the study by Coker et al. (2014) stands out for its comparative analysis of the effects of the program on different university campuses. Additionally, it included a single measurement study (Potter et al. 2011), based on the incorporation of prosocial attitude as mediating stimuli in SV cases.

*3.3. Type of Sample*

The consideration of the sample reflects the researcher's access to the population and its selection methods, the previous dissemination of the study, and its objectives (as part of informed knowledge). A non-probabilistic-voluntary sample (Cusano et al. 2020) was justified by the exposition of the research purposes and the agreement of undergraduate students interested in being part of the study.

Studies with voluntary samples (Ortiz and Shafer 2018; Cares et al. 2014; Potter et al. 2011; Martini and Piccoli 2020; Park and Kim 2021) considered a specific selection of the sample, using incentives to mobilize the interest of the participants. In some cases, by offering a symbolic financial payment or the opportunity to participate in a lottery (during the follow-up and evaluation stages).

Random probability samples equate intervention and control groups (Mennicke et al. 2018; Salazar et al. 2014, 2019; Kuffel and Katz 2002; Yount et al. 2020; Santacrose et al. 2019). All members of the population were equally likely to be selected.

In the case of a stratified random sample (Kuffel and Katz 2002), the section by strata allows its representativeness based on the population (through quotas, according to its sociodemographic conditions).

Finally, a study with purposeful sampling (Kuffel and Katz 2002) is identified based on a specific population sector, considering students residing in departments and university housing.

*3.4. Sample Size*

Regarding the size of the samples, some studies used small samples (less than 200 participants) (Cusano et al. 2020; Kuffel and Katz 2002; Martini and Piccoli 2020; Park and Kim 2021), medium samples (200–1000) (Ortiz and Shafer 2018; Salazar et al. 2014, 2019; Cares et al. 2014; Potter et al. 2008, 2011; Senn and Forrest 2016; Yount et al. 2020), and large samples (1000–15,000) (Mennicke et al. 2018; Coker et al. 2014; Santacrose et al. 2019). The demographic characteristics of the sample are as follows:

Four articles included only heterosexual or bisexual (Salazar et al. 2014, 2019; Yount et al. 2020) and homosexual (Mennicke et al. 2018) male samples; the rest included female samples, too. Some included other genders (Cusano et al. 2020; Senn and Forrest 2016) and transgender participants (Ortiz and Shafer 2018; Cares et al. 2014). Regarding age and considering that all the studies include university students, the age range reported in the majority fluctuates between 18 and 25 years, except for two studies; (Martini and Piccoli 2020) that include contracted university staff, so the range is wider (T1. $M = 44.46$ years ($SD = 9.81$), T2. $M = 45.67$ years ($SD = 7.87$), and Salazar et al. (2019), which includes a sample from 16 to 57 years old, with a mean age of 21 ($M = 21.24$; $SD = 4.68$).

Many studies collected information regarding ethnicity, which is remarkable in a country as diverse as the US (Cusano et al. 2020; Ortiz and Shafer 2018; Mennicke et al. 2018; Salazar et al. 2014, 2019; Cares et al. 2014; Potter et al. 2011; Senn and Forrest 2016; Santacrose et al. 2019). Of the studies from other countries, only one from the United Kingdom (Coker et al. 2014) collects information on this characteristic.

*3.5. Measuring Instruments*

The focus group strategy was an outcome assessment tool in the study of Cusano et al. (2020), as well as in the validation of the visual content of one of the media campaigns (Potter et al. 2008) and the qualitative identification of the outcomes perceived by participants in the "With Your Education" program (Park and Kim 2021). In these cases, qualitative analysis software is often used because it offers an exploratory and flexible research approach that provides a rich and detailed description of the study phenomena (Braun and Clarke 2006).

As expected, the three studies that implement the web-based program "RealConsent" (Gidycz et al. 2011; Salazar et al. 2014; Yount et al. 2020) share the use of specific instruments and, in turn, others depending on the purposes of the study. For example, one of them (Salazar et al. 2014) employs various measures of theoretical mediators of implied consent to assess the mechanisms through which RealConsent produces significant effects for its targets. Likewise, the Vietnamese adaptation included measures of self-efficacy and prosocial behavior of the bystander, sociodemographic data, and other control variables, as well as cognitive, attitudinal, and affective mediators and control variables aligned with the GobalConsent modules.

Social desirability is measured by two studies (Cares et al. 2014; Senn and Forrest 2016) through the Social Desirability Scale (SDS-17) (Stöber 2001). As for scales that measure variables directly related to the prosocial bystander model, we found various scales, such as the willingness to help scale (Banyard et al. 2014), bystander efficacy (Cares et al. 2014; Senn and Forrest 2016), the intention of helping a friend (Banyard et al. 2014), the choice to help a stranger (Banyard et al. 2014; Cares et al. 2014), the bystander's intention to intervene

(Banyard et al. 2005; Martini and Piccoli 2020), change/help preparation (Banyard et al. 2014), intention bystander brief (Banyard et al. 2014), barriers to intervention for sexual assault (Burn 2008), and bystander behavior adapted (Banyard et al. 2007) revised score in (Banyard et al. 2014; Senn and Forrest 2016). Another study (Santacrose et al. 2019) chose to develop a survey that grouped the measures of interest in the prosocial bystander model and the perception of the social norm of intervening in a particular situation.

Regarding myths or justifications, the Illinois rape myth acceptance scale (Payne et al. 1999) is included in the abbreviated form (Cares et al. 2014) and the original (Mennicke et al. 2018); a gender system justification scale (Jost and Kay 2005) and the updated measure to assess the subtle rape myth (McMahon and Farmer 2011) in its Italian adaptation SRMA-IT (Martini et al. 2021; Martini and Piccoli 2020).

*3.6. Study Results*

Prior to extensively commenting on the results described in Table 1, note that the heterogeneity of the programs found leads to various objectives that make it difficult to discuss the results. However, all the interventions showed positive effects in line with most of the proposed goals.

Participation in various programs was related to decreased perpetration of gender-based violence (e.g., Kuffel and Katz 2002; Coker et al. 2014; Salazar et al. 2014, 2019), as well as increases in the intentions and prosocial behaviors of viewers (e.g., Mennicke et al. 2018; Salazar et al. 2019; Cares et al. 2014; Santacrose et al. 2019; Park and Kim 2021). Although not all studies include men and women, in Santacrose et al. (2019), women were more likely to intervene in potential situations of domestic violence, intimate partner violence, sexual assault, and harassment during the 4-week follow-up.

In turn, fundamental theoretical mediators are obtained for the development and content of the program (Salazar et al. 2014). This latest study also shows that violence and bystander education for college men may benefit from including an explicit focus on decreasing harmful norms related to women.

Also included is a good adaptation of the RealConsent program included in this review (Salazar et al. 2014, 2019), successfully implemented in the United States. The Vietnamese Adaptation (GlobalConsent) (Yount et al. 2020) is the first project to adapt the program following a systematic framework and process for evidence-based program adaptation.

Likewise, the programs register progress in the identification or recognition of types of abuse (Kuffel and Katz 2002), subtle forms of violence (Martini and Piccoli 2020), knowledge of prosocial behaviors (Potter et al. 2008), a reduced adaptation of the rape myth along with less justification of the gender system (Martini and Piccoli 2020), increased awareness and sensitization of gender-based violence as a public health problem (Park and Kim 2021), understanding of the victim's perspective (Park and Kim 2021), and sexual consent (Ortiz and Shafer 2018). The former study has the particularity that college men and members of university-affiliated sororities or social fraternities obtained a more significant improvement than their respective counterparts.

Several studies did not find positive results for efficacy or perceived self-efficacy for the intervention (Salazar et al. 2014, 2019). However, perceived readiness, perceived knowledge, and self-efficacy were significantly different before and after training with another of the programs (Park and Kim 2021), whose improvements were maintained one year later. However, the same was not true for actual knowledge, following the protocol, and access to resources, which only showed significant improvement after education and significantly decreased a year later.

Finally, a qualitative study has been considered based on a data collection protocol regarding dating violence (Cusano et al. 2020). Although it does not involve pre- and post-intervention evaluation (focus group), the mere exposure of the participants to an open conversation implies the contrast of subjective presuppositions that will not maintain their original state again. Although the study does not present evidence in this regard, it opens the possibility of considering subsequent measurements to verify its effect.

## 4. Discussion

The present review intends to analyze the bystander education programs aimed at university students on their respective campuses as preventive mediation for gender-based violence. This analysis implies an assessment of the interventions, their techniques, and even their successes and limitations.

As noted in the existing literature (Cusano et al. 2020), most studies of violence on college campuses have focused on the experiences of survivors, with little emphasis on students' prior knowledge (as a preventive strategy). This fact widens the gap between the prevalence of gender-based violence and the role of bystanders (Banyard et al. 2009; Paul et al. 2013). The lack of formal university policies and the scarcity of orientation among university students favor barriers in their decision to provide help (Anderson and Danis 2007). Research allows us to analyze the bystander of gender violence as a recipient and as a manager of support for the victim.

On the other hand, in this area of intervention, the community approach stands out, which is a program that values and highlights the role of each member in the fight against gender-based violence. Its exercise is highlighted as recommended by the Centers for Disease Control and Prevention (Centers for Disease Control and Prevention 2004) in terms of public health for the prevention of violence against women since it contributes to raising awareness about the problem, provides behaviors with a modeling effect, and transmits messages and prosocial skills.

Massive interventions, known as campaigns, provided valuable information from recipients at each stage of development and were more effective [e.g., (Potter et al. 2008)], especially when counting on students to design the visual stimuli. These results highlight the need to engage key community informants and the target audience in all facets of campaign design, implementation, and execution. Knowing the bystander role encourages prosocial action in cases of gender-based violence but does not show its long-term sustainability. Simple short-term pretest/post-test design poster campaigns appear to produce an increase in willingness to intervene (Katz and Moore 2013) and are an essential strategy for the wide dissemination of preventive messages in the community (Wandersman and Florin 2003); the controversy in this regard occurs between the message received (discursive model) and the expected or actual reaction of the user, considering their motivation and previous behavior pattern; this is added to the public interest in addressing problems through communication strategies versus a humanistic model focused on the meaning of "well-being" (Werder 2017).

Promoting sensitivity towards the recognition of gender-based violence and the understanding of its categories and patterns is crucial. Society should be directed towards overcoming the perception of a "private matter" to be of public interest and, from a community health approach, providing sustainability and transcendence to the skills or actual knowledge of the participants (Park and Kim 2021), with the reduction of guilt in the victim and a defensive (protective) attitude on the part of the community (Banyard et al. 2003). However, giving help based on high efficacy and supportive intent without basic coping knowledge and skills often leads to unhelpful behaviors and adverse outcomes (Park and Ko 2020), confirming that the knowledge and skills acquired increase the intention and effectiveness when providing help. Therefore, these variables require repeated testing to ensure their potential effectiveness.

The tendency to "prevent conflicts" refers to the need to act before they manifest themselves in the stark way expected. However, it implies limited and negative connotations for coping and resolution, such as not managing or a lack of analysis and understanding of its root causes (Cascón 2001). Considering that conflict is inherent in human relations, Burton (Burton 1998) proposed the term "prevention" as the process of anticipatory intervention in a crisis. It implies an explanation and understanding of the facts, knowledge of the structural changes that allow eliminating its causes, and the promotion of collaborative attitudes that reduce the risk of future outbreaks and acts of violence.

Psychoeducational processes try to favor and provide a series of skills and strategies to better cope with conflicts in their early stages, before the crisis phase; their effectiveness depends on planned and systematic training (Cascón 2001). Therefore, most preventive interventions in gender violence (from the bystander model) include a psychoeducational structure based on awareness and reflection.

The integration of the results reflects multiple techniques of proven efficacy to mitigate gender violence in the university setting. The interventions reviewed present positive effects, but not all of them show significant effectiveness. The responsibility does not rest only on the type of intervention but may correspond to the heterogeneity of the study variables and the diversity of measurement instruments, among other non-focused variables, which agrees with what is expressed by Rubiales et al. (2018). Examples of that are non-unified criteria, instruments that are not very specific and sensitive to the construct (training variable), and the scarce replicability of the programs, among others, as an inaccurate description of the data.

This review suggests that, in the case of gender-based violence, the methodological differences between some works and others regarding the characteristics of the sample are to be trained or sensitized. This is based on the obligation to approach original research, the characterization of the sample based on the differentiated objectives, and the continuity of the participants, even more so when measuring medium- and long-term results. The term "research bias" is based on the delimitation of variables, the specification of their mediation, the conditions of the participants, their association with the study design, and the researcher's management in this regard (Manterola and Otzen 2015).

The fact that 73.3% of articles refer to interventions implemented in the United States is an essential indicator since it ratifies this country as a leader in primary research, which in the academic field allocates more funding to study strategies to curb gender-based violence. Scientific productivity depends on the national objectives and the projection of each country from the development approach and on assessing the main problems of public interest. Given the relevance of gender-based violence and its significant structural consequences, sustainable, research-based solutions must be proposed.

Adapting preventive interventions to different populations or cultural contexts is of great importance, aligning their purposes with current policies, community resources, and lines of action according to the conditions of the target population. For this purpose, each country receiving the strategy requires definition and operation in the official norms and regulations on gender-based violence (regarding the care and protection of potential victims). This condition will allow the modeling of preventive programs in various local and regional contexts. Additionally, there is the possibility that viewers have specific action procedures and institutional support networks. In this line, (Fenton and Mott 2017) makes a series of considerations to implement the bystander approach to violence prevention in Europe.

Finally, the COVID-19 health emergency and the corresponding confinement experienced in 2020 had negative consequences for witnesses of gender violence because they occurred only in the private sphere. Women only went to hospitals when the violence caused severe injuries that could not be treated independently (Nittari et al. 2021). Experiences like this warn of the need to continue fighting this scourge from other sources, such as psychoeducational and awareness-raising interventions on gender-based violence that involve society and reinforce support resources for women victims, as well as access to care and protection against a possible condition of violence.

*Limitations and Future Directions*

This study is not without limitations; they are shown as follows: (a) consider only six specialized databases; (b) the use of terms that could be insufficient for the identification of a more significant number of articles; (c) the lack of analysis of the characteristics of each program in its methodology and activities; and (d) the role of the intervention in the assimilation of prosocial behaviors in people who perpetrate gender-based violence.

In addition, some of the studies analyzed do not report the sampling strategies used or the type of design they followed. There is no inclusion of gray literature with the use of secondary databases such as Google Scholar. They may be overcome in future research.

Although a risk assessment is not required in a scoping review, it would be interesting to include such an assessment in a future systematic review.

The use of terms symbolized primary difficulty in searching for articles in the databases because, although the objective was to find interventions towards the modification of beliefs, attitudes, intentions, or behaviors of the viewers, there was some controversy with the term "intervention," since most of the articles were based on the theory of the bystander model and their actions in potential situations of gender-based violence. In this way, we identified numerous descriptive and non-exploratory articles on the cross-sectional and diagnostic measurement of the perceptive barriers of the participants, which has reduced the selection of studies. In turn, there are numerous concepts that refer to violence against women in English literature, like "violence against women," "gender-based violence," "intimate partner violence," "dating violence," and "domestic violence". Besides, specific terminology the type of violence is added, such as "sexual violence," which leads or directs the meaning of the interventions, adding to the fact that the databases have not incorporated all the related terms, a situation that contributed to a bias in the inclusion of all existing studies in the field. Despite this, the fact that other articles in high-impact journals have used this combination of terms is essential to note, considering the possibility that some psychological indicators are omitted.

Finally, the vast heterogeneity of variables and instruments used in the different studies should be considered, as it hindered a joint analysis of the interventions and, therefore, the development of generalizations.

## 5. Conclusions

The 15 articles included in this review are based on informative and awareness programs for university students for the assimilation and execution of prosocial behaviors in response to acts of gender violence. These studies have an experimental design: one of qualitative measurement (Cusano et al. 2020) and 14 of quantitative measurement, among these four experimental studies (Kuffel and Katz 2002; Salazar et al. 2014; Yount et al. 2020; Santacrose et al. 2019), three quasi-experimental studies (Coker et al. 2011; Caprara et al. 2005; Kuffel and Katz 2002), six pre-experimental studies (Ortiz and Shafer 2018; Mennicke et al. 2018; Coker et al. 2014; Senn and Forrest 2016; Martini and Piccoli 2020; Park and Kim 2021), and one single measurement study of reaction to mediating stimuli (Potter et al. 2011).

According to the type of sample, one case of a sample interested in the purpose of the research (Cusano et al. 2020), six cases of a sample promoted by external incentives (Ortiz and Shafer 2018; Cares et al. 2014; Potter et al. 2011; Senn and Forrest 2016; Martini and Piccoli 2020; Park and Kim 2021), six cases of a probability sample (Mennicke et al. 2018; Salazar et al. 2014, 2019; Kuffel and Katz 2002; Yount et al. 2020; Santacrose et al. 2019), one stratified random sample (Kuffel and Katz 2002), and one purposive sample (Potter et al. 2008) were identified. Regarding sample size, four studies with small samples (<200 participants), eight studies with median samples (200–1000 participants), and three studies with large samples (1000–15,000 participants) were identified.

The sociodemographic characteristics of the sample included gender diversity and sexual choice. The most frequent age range was 18 to 25 years for university students and up to 50 years for administrative personnel. Additionally, 60% of the studies presented information on the ethnic origin of the users.

The measurement instruments include semi-structured interviews, scales, and questionnaires based on consent and sexual self-efficacy, prosocial behavior, social desirability, and intention to help. Additionally, there are some cognitive, attitudinal, and affective mediators and, finally, justification of SA, myths about SA, and gender stigmas. All the studies

obtained positive results. However, it is not feasible to standardize their effectiveness given the methodological differences in each study.

Finally, it is suggested that the communication of these results and other research imply a contribution to prosocial and proactive intervention on gender violence by promoting and implementing equitable and intercultural public policies based on educational programs of proven effectiveness in favor of women and vulnerable minority groups.

**Author Contributions:** All authors contributed equally to this review. All authors have read and agreed to the published version of the manuscript.

**Funding:** This research received no external funding.

**Institutional Review Board Statement:** Not applicable.

**Informed Consent Statement:** Not applicable.

**Data Availability Statement:** The analysis script is available upon request from the authors.

**Acknowledgments:** The authors thank the casual helpers in information processing and search.

**Conflicts of Interest:** The authors declare no conflict of interest.

## Appendix A

**Table A1.** Preferred Reporting Items for Systematic reviews and Meta-Analyses extension for Scoping Reviews (PRISMA-ScR) Checklist.

| Section | Item | PRISMA-ScR Checklist Item | Reported on Page |
|---|---|---|---|
| | | Title | |
| Title | 1 | Identify the report as a scoping review. | 1 |
| | | Abstract | |
| Structured summary | 2 | Provide a structured summary that includes (as applicable): background, objectives, eligibility criteria, sources of evidence, charting methods, results, and conclusions that relate to the review questions and objectives. | 1 |
| | | Introduction | |
| Rationale | 3 | Describe the rationale for the review in the context of what is already known. Explain why the review questions/objectives lend themselves to a scoping review approach. | 1–3 |
| Objectives | 4 | Provide an explicit statement of the questions and objectives being addressed regarding their key elements (e.g., population or participants, concepts, and context) or other relevant key elements used to conceptualize the review questions and/or objectives. | 3 |
| | | Methods | |
| Protocol and registration | 5 | Indicate whether a review protocol exists; state if and where it can be accessed (e.g., a Web address); and if available, provide registration information, including the registration number. | 4 |
| Eligibility criteria | 6 | Specify characteristics of the sources of evidence used as eligibility criteria (e.g., years considered, language, and publication status), and provide a rationale. | 4 |

**Table A1.** *Cont.*

| Section | Item | PRISMA-ScR Checklist Item | Reported on Page |
|---|---|---|---|
| Information sources | 7 | Describe all information sources in the search (e.g., databases with dates of coverage and contact with authors to identify additional sources), as well as the date the most recent search was executed. | 5 |
| Search | 8 | Present the full electronic search strategy for at least 1 database, including any limits used, such that it could be repeated. | 5 |
| Selection of sources of evidence | 9 | State the process for selecting sources of evidence (i.e., screening and eligibility) included in the scoping review. | 5–6 |
| Data charting process | 10 | Describe the methods of charting data from the included sources of evidence (e.g., calibrated forms or forms that have been tested by the team before their use, and whether data charting was done independently or in duplicate) and any processes for obtaining and confirming data from investigators. | 6 |
| Data items | 11 | List and define all variables for which data were sought and any assumptions and simplifications made. | 5 |
| Critical appraisal of individual sources of evidence | 12 | If done, provide a rationale for conducting a critical appraisal of included sources of evidence; describe the methods used and how this information was used in any data synthesis (if appropriate). | N.A. |
| Synthesis of results | 13 | Describe the methods of handling and summarizing the data that were charted. | 3–4 |
| Results | | | |
| Selection of sources of evidence | 14 | Give number of sources of evidence screened, assessed for eligibility, and included in the review, with reasons for exclusions at each stage, ideally using a flow diagram. | 6 |
| Characteristics of sources of evidence | 15 | For each source of evidence, present characteristics for which data were charted and provide the citations. | 8–12 |
| Critical appraisal within sources of evidence | 16 | If done, present data on critical appraisal of included sources of evidence (see item 12). | N.A. |
| Results of individual sources of evidence | 17 | For each included source of evidence, present the relevant data that were charted that relate to the review questions and objectives. | 2–4 |
| Synthesis of results | 18 | Summarize and/or present the charting results as they relate to the review questions and objectives. | 4-5 |
| Discussion | | | |
| Summary of evidence | 19 | Summarize the main results (including an overview of concepts, themes, and types of evidence available), link to the review questions and objectives, and consider the relevance to key groups. | 5-6 |
| Limitations | 20 | Discuss the limitations of the scoping review process. | 7 |
| Conclusions | 21 | Provide a general interpretation of the results with respect to the review questions and objectives, as well as potential implications and/or next steps. | 8 |
| Funding | | | |
| Funding | 22 | Describe sources of funding for the included sources of evidence, as well as sources of funding for the scoping review. Describe the role of the funders of the scoping review. | 9 |

Note. N.A.: Not applicable.

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
