# Peer review of "A Scoping Review of Educational Interventions to Increase Prosociality against Gender-Based Violence in University Bystanders"

_socsci, doi:10.3390/socsci12070406_

Round 1
Reviewer 1 Report
The paragraph under 2.2. search, the author mentions the research which was carried out in August and September 2021. stating that it was done "in all different and cultural contexts registering studies from all countries.......". Here the author must be specific by mentioning particular countries within which the research was conducted. This is on page 5/24
Author Response
The paragraph under 2.2. search, the author mentions the research which was carried out in August and September 2021. stating that it was done "in all different and cultural contexts registering studies from all countries.......". Here the author must be specific by mentioning particular countries within which the research was conducted. This is on page 5/24
Thank you for your comment. We have included the countries in which the research was conducted in the 2.2 section.

Reviewer 2 Report
The paper largely reads well and certainly makes useful contribution to debate.
Some sentences require further explication. For example:
Page 2 second main paragraph (line 12 from the top) beginning ' What allows to generalise...' This sentence is hard to follows as it stands.
The language is quite emotive in places...Under 1.2 page 2 second paragraph 'suffered sexual assault' rather than 'experienced sexual assault'. Indeed, some judicial editing is required to ensure that commas are used instead of semi-colons where appropriate.
The punctuation is rather odd in places. For example, page 2 third complete paragraph 'In addition to the approach to diffusion of responsibility, where bystanders do not intervene in the presence of others who could intervene [6]; it is theorised....' I would suggest a comma rather than a semi-colon after [6].
Page 4 2.1 Protocol - evidences 'a' systematic review, rather than 'the' systematic review.
The inclusion and exclusion criteria are explained well.
The article is interesting in substance, but some editorial attention is required. Also, there is a tendency to write in the passive tense e.g. page 12 'In this regard, a study is identified that evaluates students' experiences, behaviours, and attitudes...' The writing could therefore be livelier and more engaging. At present, it is written as if writing up a lab experiment.
Page 15 - under 4. Discussion, why is their reference to 'This study intends to analyse'.... The present study analyses surely. 'Said analysis implies....' 'The' said analysis...?
The limitations are well drawn.
Interesting overall, and the paper has merit, but it is not easy to read in places. Think of the reader more when conveying different points.
Author Response
Comments and Suggestions for Authors
The paper largely reads well and certainly makes useful contribution to debate.
Some sentences require further explication. For example:
Page 2 second main paragraph (line 12 from the top) beginning ' What allows to generalise...' This sentence is hard to follows as it stands.
Thank you for your comment, we have changed the sentence to facilitate the reading.
The language is quite emotive in places...Under 1.2 page 2 second paragraph 'suffered sexual assault' rather than 'experienced sexual assault'. Indeed, some judicial editing is required to ensure that commas are used instead of semi-colons where appropriate.
Both the expression and the judicial editing have been modified, thank you for the tip.
The punctuation is rather odd in places. For example, page 2 third complete paragraph 'In addition to the approach to diffusion of responsibility, where bystanders do not intervene in the presence of others who could intervene [6]; it is theorised....' I would suggest a comma rather than a semi-colon after [6].
Punctuation has been improved in the whole document.
Page 4 2.1 Protocol - evidences 'a' systematic review, rather than 'the' systematic review.
We have change it following your recommendations, thank you.
The inclusion and exclusion criteria are explained well.
We really appreciate this comment; it helps us to continue with our work.
The article is interesting in substance, but some editorial attention is required. Also, there is a tendency to write in the passive tense e.g. page 12 'In this regard, a study is identified that evaluates students' experiences, behaviours, and attitudes...' The writing could therefore be livelier and more engaging. At present, it is written as if writing up a lab experiment.
We acknowledge your recommendation and we have change this sentence, we hope it has been improved. Thank you for your appreciation.
Page 15 - under 4. Discussion, why is their reference to 'This study intends to analyse'.... The present study analyses surely. 'Said analysis implies....' 'The' said analysis...?
Thank you for this observation. In the modified manuscript, we have taken your comment into account.
The limitations are well drawn.
Thank you for this comment.
Interesting overall, and the paper has merit, but it is not easy to read in places. Think of the reader more when conveying different points.
We highly appreciate your time, your comments and your recommendations, we will make sure we follow all of them. We have reviewed the entire manuscript, and believe that the document has been improved by your comments. Thank you.

Reviewer 3 Report
Dear authors,
Congratulations on your article! Overall, it is well-structured and provides a clear overview of literature in the field. The methods section follows the PRISMA protocol, and the data is available, which is commendable. I find the article to be well-written, clear, and concise.
However, I have a couple of suggestions to improve it further. Firstly, it would be helpful to improve the cohesion of the introduction section, making it clearer and more focused. Secondly, it would be beneficial to add some implications for policy-making at the end of the article. This could provide valuable information for a relevant topic and make your article even more impactful.
Thank you for your work, and I look forward to seeing the revised version.
Best regards,
Reviewer.
Overall English is fine, the introduction section could be improve. But not a mejor concern.
Author Response
Dear authors,
Congratulations on your article! Overall, it is well-structured and provides a clear overview of literature in the field. The methods section follows the PRISMA protocol, and the data is available, which is commendable. I find the article to be well-written, clear, and concise.
Thank you very much for your comments.
However, I have a couple of suggestions to improve it further. Firstly, it would be helpful to improve the cohesion of the introduction section, making it clearer and more focused. Secondly, it would be beneficial to add some implications for policy-making at the end of the article. This could provide valuable information for a relevant topic and make your article even more impactful.
Thank you for this observation. We have revisited the introduction section and improved its cohesion. Also, we have included policy-making implications at the discussion section. Also, we have edited the English language.
Thank you for your work, and I look forward to seeing the revised version.
We appreciate your time and your recommendations. Thank you.
Best regards,

Reviewer 4 Report
Title: A systematic review of educational interventions to increase prosociality against gender-based violence in university bystanders
General comment
The aim of this paper would be to analyze preventive interventions based on gender-based violence towards university viewers as a basis for future proven and sensitive implementation processes of intercultural adaptation; but there are many problems both in terms of content and at the methodological level (very serious problems).A systematic review requires a lot of effort and detailed knowledge of the methodology. If you realize that it is too complex it is better to resort to other types such as a scoping review or by further downgrading a narrative review.
Specific comments
The introduction is excessively vague and poorly focused on the purpose of the study. When we talk about gender, we should first of all talk about the Social determinants of health. In this regard it is suggested to analyze and include the following article: https://doi.org/10.13136/isr.v13i1.589.
The following article is recommended on the importance of workplace/academic violence: https://doi.org/10.1002/(SICI)1098-2337(1996)22:3<161::AID-AB1>3.0.CO;2-Q.
Furthermore, in any type of review essential is the question research which is currently not present, and this already makes the work very scarce from a methodological point of view.
The introduction is too long, it should be summarized more.
Material and methods: This section is seriously lacking: 1) there is no bias risk assessment which is mandatory if a systematic review is adopted, while it is not mandatory if a scoping review is used. Eventually the authors can adopt this typology. The evaluation of the inclusion and exclusion criteria should be made more methodological, for these reasons it is requested to introduce PICOTS.
A very serious shortcoming concerns the total absence of inclusion of gray literature with the use of secondary databases such as Google Scholar. Provision should be made to include this search with new items included.
A stistematic review CANNOT be based on only 15 articles. Such a low number can be used for a Rapid review which, like scoping, does not require the risk assessment of the bias. The role of the various review team members was not specified. It is necessary to do this.
The results cannot be evaluated due to issues related to methodological shortcomings and inclusion of new articles. The table is too discursive, its contents must be broadly summarized.
The discussion is excessively long and vague. There is no reference to any comparison between the results obtained and the available literature. No form of strategy is reported for solving the problem, and usually in literature reviews resolution strategies are recommended based on the research questions formulated.
The conclusions are too general, vague and broad. I suggest a broad synthesis of the conclusions as well.
An extensive revision of the language is needed
Author Response
Title: A systematic review of educational interventions to increase prosociality against gender-based violence in university bystanders
General comment
The aim of this paper would be to analyze preventive interventions based on gender-based violence towards university viewers as a basis for future proven and sensitive implementation processes of intercultural adaptation; but there are many problems both in terms of content and at the methodological level (very serious problems).A systematic review requires a lot of effort and detailed knowledge of the methodology. If you realize that it is too complex it is better to resort to other types such as a scoping review or by further downgrading a narrative review.
We acknowledge the reviewer comment; we have made the pertinent changes to be considered a Scoping review thanks to the suggestions made.
Specific comments
The introduction is excessively vague and poorly focused on the purpose of the study. When we talk about gender, we should first of all talk about the Social determinants of health. In this regard it is suggested to analyze and include the following article: https://doi.org/10.13136/isr.v13i1.589.
The following article is recommended on the importance of workplace/academic violence: https://doi.org/10.1002/(SICI)1098-2337(1996)22:3<161::AID-AB1>3.0.CO;2-Q.
Thank you for your observation. On the other hand, due to the comments of 3 reviewers of the present manuscript, we are not yet able to identify which part of the manuscript is “excessively vague and poorly focused on the purpose of the study”. However, we have included both articles and improved the introduction section.
Furthermore, in any type of review essential is the question research which is currently not present, and this already makes the work very scarce from a methodological point of view.
Thank you for your observation. We have included the research question, thank you for pointing it out.
The introduction is too long, it should be summarized more.
Thank you for your observation. We have followed your recommendation and shortened the introduction.
Material and methods: This section is seriously lacking: 1) there is no bias risk assessment which is mandatory if a systematic review is adopted, while it is not mandatory if a scoping review is used. Eventually the authors can adopt this typology. The evaluation of the inclusion and exclusion criteria should be made more methodological, for these reasons it is requested to introduce PICOTS.
We agree with your comment, and we have changed the type of research to a Scoping review due to the absence of a bias risk assessment. Moreover, we have included the following PICO question: The initial question (PICO) that guided this review was as follows: Do the educational interventions increase prosociality in the face of gender violence in university bystanders?
A very serious shortcoming concerns the total absence of inclusion of gray literature with the use of secondary databases such as Google Scholar. Provision should be made to include this search with new items included.
Thanks to changing into a Scoping Review, we have included lack of grey literature as an exclusion criterium. Also, it has been considered as a limitation in our work.
A stistematic review CANNOT be based on only 15 articles. Such a low number can be used for a Rapid review which, like scoping, does not require the risk assessment of the bias. The role of the various review team members was not specified. It is necessary to do this.
We acknowledge your comment. As the article is now considered as a scoping review, the bias risk assessment is no longer mandatory. Also, the role of the team members is now included in the article. Thanks for pointing that out.
The results cannot be evaluated due to issues related to methodological shortcomings and inclusion of new articles. The table is too discursive, its contents must be broadly summarized.
The discussion is excessively long and vague. There is no reference to any comparison between the results obtained and the available literature. No form of strategy is reported for solving the problem, and usually in literature reviews resolution strategies are recommended based on the research questions formulated.
The conclusions are too general, vague and broad. I suggest a broad synthesis of the conclusions as well.
Content has been extensively reviewed and improved by the team members. Also, it has been summarized in the table of results.
We highly appreciate your time and comments to improve our article, it is an opportunity for us to continue improving. Thank you.
Comments on the Quality of English Language
An extensive revision of the language is needed
English language has been deeply reviewed and improved.

Round 2
Reviewer 4 Report
The revised paper is well structured and I am satisfied with how the authors followed my suggestions.
Good job!